# Local area public sector spending and nutritional anaemia hospital admissions in England: a longitudinal ecological study

Rosemary Jenkins [1], Eszter P Vamos,[1] Kate E Mason [2], Konstantinos Daras [2], David Taylor-Robinson,[2] Clare Bambra,[3] Christopher Millett,[1] Anthony A Laverty [1]

[1]Public Health Policy Evaluation Unit, School of Public Health, Imperial College London, Charing Cross Campus; The Reynolds Building; St Dunstan's Road, London W6 8RP, UK
[2]Department of Public Health, Policy and Systems, Institute of Population Health, University of Liverpool; Waterhouse Building Block F, 2nd Floor, Liverpool L69 3BX, UK
[3]Population Health Sciences Institute, Faculty of Medical Sciences, Newcastle University, NE1 4LE, UK

**Correspondence to**
Dr Anthony A Laverty;
a.laverty@imperial.ac.uk

## ABSTRACT

**Introduction** Reductions in local government spending may have impacts on diets and health which increase the risk of hospital admissions for nutritional anaemias. Mechanisms include potential impacts of changes to local authority (LA) services (eg, housing services) on personal resources and food access, availability and provision. We therefore investigated the association between changes in LA spending and nutritional anaemia-related hospital admissions. Specifically, we address whether greater cuts to LA spending were linked to increased hospital admissions for nutritional anaemias.

**Design** Longitudinal analysis of LA panel data using Poisson fixed effects regression models.

**Setting** 312 LAs in England (2005–2018).

**Main exposure** Total LA service expenditure per capita per year.

**Main outcome** Principal and total nutritional anaemia hospital admissions, for all ages and stratified by age (0–14, 15–64, 65+ years).

**Results** LA service expenditure increased by 9% between 2005 and 2009 then decreased by 20% between 2010 and 2018. Total nutritional anaemia hospital admissions increased between 2005 and 2018 from 173 to 633 admissions per 100 000 population. A £100 higher LA service spending was associated with a 1.9% decrease in total nutritional anaemia hospital admissions (adjusted incidence rate ratio (aIRR): 0.98, 95% CI: 0.96 to 0.99). When stratified by age, this was seen only in adults. A £100 higher LA service spending was associated with a 2.6% decrease in total nutritional anaemia hospital admissions in the most deprived LAs (aIRR: 0.97, 95% CI: 0.95 to 1.0).

**Conclusion** Increased LA spending was associated with reduced hospital admissions for nutritional anaemia. Austerity-related reductions had the opposite effect, increasing admissions, with greater impacts in more deprived areas. This adds further evidence to the potential negative impacts of austerity policies on health and health inequalities. Among other impacts, re-investing in LA services may prevent hospital admissions associated with nutritional anaemia.

## STRENGTHS AND LIMITATIONS OF THIS STUDY

⇒ This study uses national data to assess relationships between public sector spending and hospital admissions.
⇒ A fixed effect panel design allows us to examine within local authority changes in both exposures and outcomes.
⇒ However, we examined hospitalisations only, which does not give a full picture of burden of disease, as measures such as general practice diagnoses would.
⇒ The natural experimental design means that we cannot infer causality and research at the individual level would assist with this in the future.

## INTRODUCTION

Nutritional anaemias related to poor diet are an important public health issue, increasing in prevalence over recent years. Nutritional anaemias involve reductions in red blood cells or their haemoglobin content due to a low nutritional intake of certain micronutrients, particularly iron, vitamin $B_{12}$ and folate.[1] Nutritional anaemias are associated with a range of adverse health outcomes—for example, iron deficiency anaemia can lead to poor outcomes in those with coronary heart disease.[2] Furthermore, nutrient deficiencies, which cause nutritional anaemias, are important risk factors for diseases through multiple physiological mechanisms.[3 4] For example, hospitalisations for iron deficiency anaemias increased from 67 592 in 2008–2009 to 131 064 in 2016–2017.[5] A 2019 report by the UK House of Commons Environmental Audit Committee described the presence of a double burden of malnutrition in the UK, with overweight and obesity coexisting with undernutrition, both leading to micronutrient deficiencies.[6]

The increase in hospital admissions for nutritional anaemias has occurred at the same time as large cuts to public services in the UK.[5] In 2010, the UK government introduced austerity policies which led to considerable changes to the UK benefits system and

reductions in local authority (LA) funding and service expenditure.[7 8] Reductions in LA funding had greater impacts in urban and more deprived councils, especially in the north of England, leading to differential impacts on the wide range of services that LAs provide.[9–11] These services include social care, housing, highways and transport, environment and regulatory and planning and development services.[11] Systematic review evidence demonstrates that these services may be important in improving the social determinants of health and health inequalities.[12] Research reports have found that reductions to LA service spending may have had negative impacts on health and widened health inequalities.[13] For example, a £100 decrease in annual LA funding per person has been associated with a decrease in life expectancy at birth of 1.2–1.3 months between 2013 and 2017.[14]

The relationship between changes in local government spending and risk of nutritional anaemia is unclear. Much of the extant research on nutritional impacts of austerity policies in the UK has focused on foodbank use and not other potential outcomes such as hospital admissions.[15] Risk of nutritional anaemias caused by poor diets may be plausibly impacted by changes to public sector spending through a range of different mechanisms. Decreases in adult social care expenditure, particularly in more deprived LAs, may have led to fewer residential and domiciliary care placements, which could have precipitated changes in food provision to vulnerable people if they are not able to access the food provided through such care placements.[16] Decreases in Meals on Wheels services, lunch clubs and other food provision may also have a similar effect.[17 18] These services have been shown to be important in providing vulnerable adults with the calories and nutrients that they need, preventing malnutrition.[19 20] Thus, decreases in these services may directly lead to increases in nutritional anaemias in these individuals due to deficits in nutrients. Fewer carers visiting homes may also lead to issues with food acquisition as individuals may rely on carers for shopping, which may also lead to changes in food intakes and potential nutritional anaemias. Changes to LA environmental and regulatory spending and planning and development spending may influence access and availability of foods through changes to local businesses, hygiene inspections and food regulation, which may also lead to changes in food purchasing, particularly takeaways.[21]

There are also pathways which may indirectly lead to reductions in micronutrient consumption and nutritional anaemias. Through factors including less housing advice and support, less money spent on housing renewal, poorer quality homes and lower spending on homelessness, reductions in LA housing service spending may lead to trade-offs between housing costs and paying for other items, including healthy food, due to a reduction in personal resources.[22 23] As micronutrient-rich foods such as fruits and vegetables tend to be more expensive than ultra-processed foods which are generally low in micronutrients, these changes in resources may indirectly lead to nutritional anaemias through changes in types of foods purchased and consumed.[24] Lower LA highways and transport spending may have similar impacts on personal resources and therefore food purchasing due to increased costs of public transport.[11 25 26] Decreases in highways and transport spending may also lead to reductions in public transport (eg, number of different routes, number of buses per day).[11 25 26] This may adversely affect individuals' access to supermarkets, potentially leading to decreases in nutritious foods, increases in ultra-processed foods and increases in takeaways.[27 28] Finally, a decrease in money spent on cultural services (including recreation and sport, library services, culture and heritage and open spaces) may indirectly lead to micronutrient depletion through changes to lifestyle (such as physical activity) and potential increases in stress and mental health issues, though further research is needed in this area.[21 29] Thus, there are a number of potential pathways which may influence diagnoses of nutritional anaemias (figure 1). We therefore aimed to investigate the association between changes to LA service spending and principal and total nutritional anaemia hospital admissions in LAs in England. Given the role of LAs in addressing the social determinants of health, we hypothesise that effects may be stronger in more deprived LAs and those experiencing greater reductions in working age benefits.

## METHODS
### Data
#### Exposure variables
The time period for our analysis was 2005–2018. Annual LA service expenditure is reported by the Ministry of Housing, Communities and Local Government, compiled and freely available from the Place-based Longitudinal Data Resource (PLDR) for 326 LAs.[30–36] These data are based on financial years rather than calendar years. We used net expenditure data for lower tier LAs, which excludes income from services. The Islands of Scilly and the City of London were not included due to small populations, and an additional 12 LAs were excluded due to missing covariate data. We calculated total net LA expenditure excluding police services and fire and rescue services (as these are funded by separate grants and precepts and provided by separate authorities), education services (due to changes in schooling provision over the study period), public health services (due to data not being available for all years due to changes in LA responsibilities) and court services (due to data not being available for all years).[37] All data were adjusted for inflation using the Consumer Price Index with 2015 as the reference year.[38]

#### Outcome variables
Hospital Episode Statistics (HES) is a database of all National Health Service (NHS) hospitals' admissions, Accident and Emergency attendances and outpatient

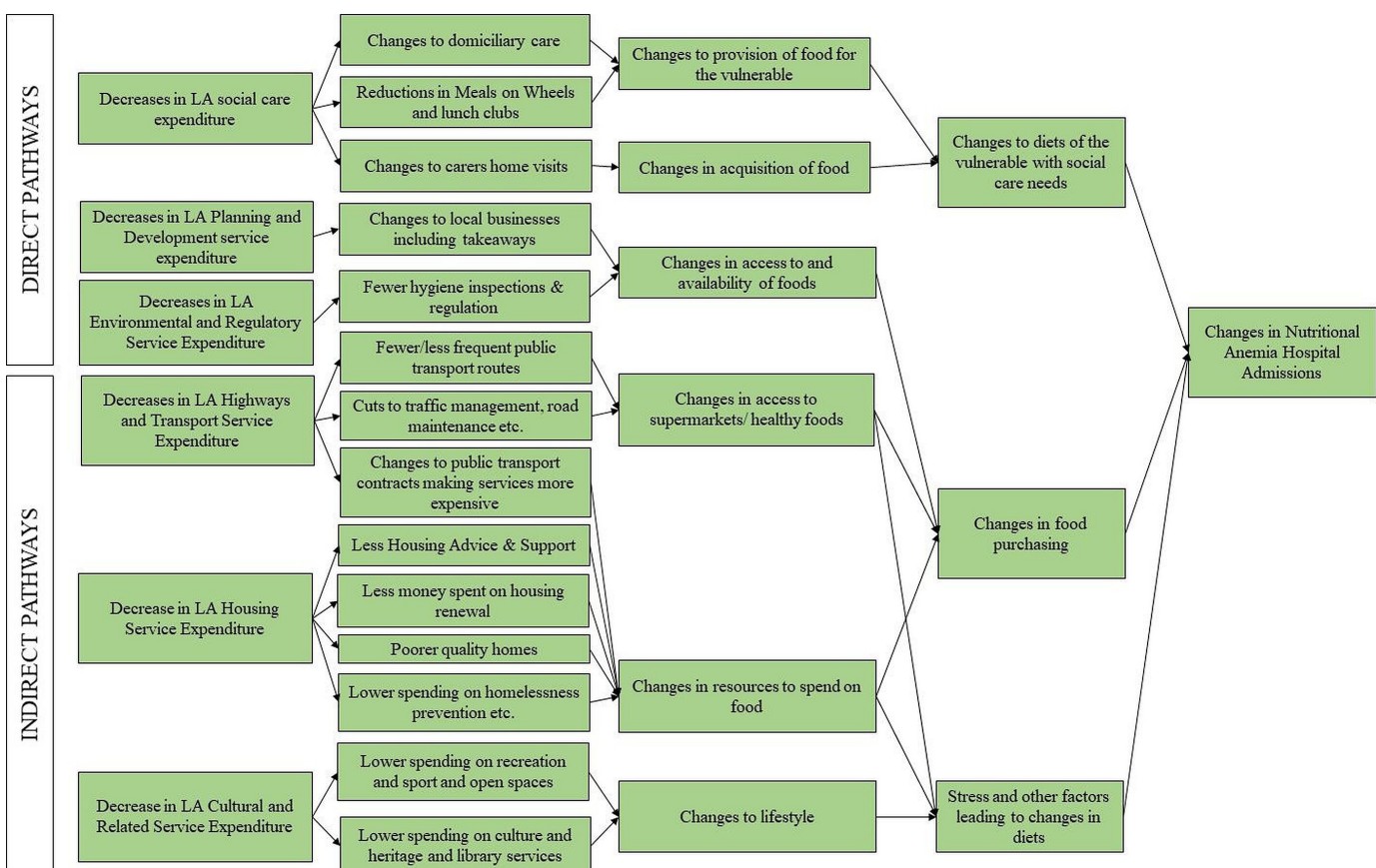

**Figure 1** Logic model of potential pathways through which changes to LA spending may affect nutritional anaemias. LA, local authority.

appointments in England.[39] For this study, we used yearly LA-level counts of inpatient or day-case admissions of nutritional anaemias in NHS hospitals available as safeguarded data sets from the PLDR.[40–43] These were iron deficiency anaemia (excluding iron deficiency anaemia due to blood loss), vitamin $B_{12}$ deficiency anaemia, folate deficiency anaemia and other nutritional anaemias by LA. We obtained principal nutritional anaemia hospital admissions data where the nutritional anaemia was the 'condition established after study to be chiefly responsible for occasioning the admission of the patient to the hospital for care'.[44] We also obtained data for hospital admissions where nutritional anaemias are a secondary diagnosis, an additional condition that an individual may have during a hospital admission for a different principal diagnosis. These were given by each age (0–14, 15–64, 65+) or sex group individually. We aggregated these data to obtain principal nutritional anaemia hospital admissions and total nutritional anaemia hospital admissions (consisting of both principal and secondary diagnoses). In this paper, we focus on total nutritional anaemia hospital admissions as they key outcome for a number of reasons, including (1) principal nutritional anaemia hospital admissions are relatively uncommon, and thus total nutritional anaemia hospital admissions are more reflective of the burden of nutritional anaemias in the

population and (2) the multisystem impact of nutritional anaemias means that the primary diagnoses may be considered to be a manifestation of nutritional anaemias as they led to the patient being ill enough to seek medical care. We used annual LA population estimates from the Office for National Statistics (ONS) to generate admission rates per 100 000 population.[45]

For sensitivity analyses, we also obtained counts of nutritional anaemia admissions excluding those alongside a principal or secondary diagnosis of another condition that may cause nutritional deficiencies, such as cachexia, malabsorption, malignancies and diabetes (full list in online supplemental appendix 1). As with the main outcome variable above, these data were obtained stratified by age or sex. These data were also aggregated to obtain principal and total (ie, principal and secondary) nutritional anaemia hospital admissions.

In the regression models we adjusted for the proportion of the LA population that is working age, proportion of LA population that is male, gross disposable household income (GDHI) and unemployment rate. The proportion of LA population that was working age (aged 15–64) and male were calculated from ONS data.[45] These covariates were included to account for differences in changing LA demographics. We obtained GDHI data, an area-level measure of individuals'

available money for spending or saving following payment of taxes and receipt of benefits, adjusted for inflation, from the ONS.[46] LA unemployment rates were also obtained from the ONS, who calculate them based on a model using estimates from the Labour Force Survey and counts of people claiming unemployment benefits.[47] GDHI and unemployment rate were used as markers of available household resources at the LA level. Time invariant factors were not adjusted for as they were removed by the fixed effects model, however, we also obtained Index of Multiple Deprivation (IMD) and level of reductions in working-age benefits as they were identified a priori as potential effect modifiers. IMD 2015 was obtained from the Ministry of Housing, Communities and Local Government.[48] We used the population-weighted rank of average rank of LA IMD to make quintiles of relative deprivation. We also used the Uneven Impact of Welfare Reform data set (provided by its authors) which estimated the cumulative reductions in working-age people's benefits for each LA due to welfare reforms since 2010.[49] Previous research has found strong associations between reductions in welfare benefits and clinical outcomes.[50] We derived quartiles of this variable to assess level of reductions to working-age benefits as a potential effect modifier.

## Analyses

We descriptively examined total LA service expenditure in 2005, 2010 and 2018, including differences by covariates and proposed effect modifiers. We also calculated percentage changes before (2005–2009) and after (2010–2018) the introduction of austerity measures. We tabulated mean principal and total nutritional anaemia hospital admissions for the full sample and stratified by age group, presented as rates per 100 000 population, and plotted their change over time. We also plotted change over time stratified by sex.

We used a Poisson fixed effects panel regression approach for the main analysis. Panel regression models can be used to investigate LA units over time while taking data clustering over time into account.[51] Poisson fixed effects regression was used as a robust approach to modelling count panel data.[52] We used cluster robust SEs to adjust for potential autocorrelation and heteroskedasticity.[53] We also created dummy variables for each year and used them to account for England-wide time effects. LA population was used as an offset variable to account for population size. For our primary analysis, we undertook a Poisson fixed effects analysis to investigate the relationship between total LA service spending and total nutritional anaemia hospital admissions. We present adjusted models (with covariates as proportion of the LA population that is working age, proportion of the LA population that is male, GDHI and unemployment rate). We also subsequently stratified by IMD quintiles and level of benefit reductions quartiles to test a priori hypotheses

regarding effect modification. We undertook analyses stratified by age and sex. We separately analysed principal nutritional anaemia hospital admissions as the outcome, with adjusted and stratified Poisson models as above.

We also undertook two sensitivity analyses. In the first, we used as the outcome variable total nutritional anaemia hospital admissions excluding admissions alongside a diagnosis of another condition which may cause a nutritional deficiency. While public sector spending reductions could cause or exacerbate this wider set of conditions, this sensitivity analysis focuses on diagnoses of being admitted *for* nutritional anaemia rather than *with* it. In the second sensitivity analysis, we excluded social care expenditure from the exposure variable. Unlike most areas of LA service expenditure, some LAs had to increase their social care expenditure after 2010 due to demand. Thus, LAs with higher spending on social care may have higher levels of nutritional anaemias, which may lead to issues of reverse causality when social care is included in the exposure variable.[54] Our sensitivity analysis investigated the relationship between LA spending excluding social care and total nutritional anaemia hospital admissions.

### Patient and public involvement

Plans for this research were presented to a committee of three public experts in November 2020. The public experts provided feedback on the research plans and appropriate changes were made following the meeting.

## RESULTS

On average, LA service spending increased by 9% between 2005 and 2009, and then decreased by 20% between 2010 and 2018 (table 1). Mean total LA spending per year is plotted in online supplemental appendix 2. Between 2005 and 2009, LAs in Yorkshire and the Humber (16%), the East Midlands (15%) and the North West (14%) increased expenditure the most, while London (4%) and the South East (5%) increased expenditure the least. However, LAs with the highest unemployment rates decreased expenditure between 2005 and 2009. Following 2010, London had the greatest decrease of 31%, followed by the Northern regions (a 27% decrease for the North East and 21% for both the North West and Yorkshire and the Humber). More deprived LAs experienced greater cuts (28% in the most deprived quintile compared with 12% for the least deprived quintile). Greater reductions were also seen in areas with higher unemployment rates and larger reductions to working age benefits.

Total nutritional anaemia hospital admissions have increased since 2005, from 173 per 100 000 population in 2005 to 632 per 100 000 population in 2018 (figure 2). Total nutritional anaemia hospital admissions in 65+ adults increased between 2005 and 2010, with the rate of increase slowing between 2010 and

**Table 1** Mean LA service expenditure in 2005, 2010 and 2018 and percentage change, stratified by socio-demographic variables

| | Total per capita LA spending 2005 | Total per capita LA spending 2010 | Total per capita LA spending 2018 | Change 2005–2009 | Change 2010–2018 | Change 2005–2018 |
|---|---|---|---|---|---|---|
| Total | 845.4 (237.4) | 861.6 (202.7) | 690.9 (103.4) | +9.2 | −19.8 | −18.3 |
| Region | | | | | | |
| North East | 1007.0 (153.9) | 1060.7 (166.8) | 776.7 (78.6) | +11.5 | −26.8 | −22.9 |
| North West | 882.7 (192.5) | 943.9 (187.0) | 741.7 (82.9) | +14.4 | −21.4 | −16.0 |
| Yorkshire and the Humber | 773.1 (106.5) | 839.5 (119.4) | 661.7 (77.1) | +16.1 | −21.2 | −14.4 |
| East Midlands | 705.0 (64.3) | 776.7 (111.0) | 620.0 (72.0) | +14.5 | −20.1 | −12.1 |
| West Midlands | 770.2 (98.7) | 768.7 (120.3) | 655.8 (73.1) | +5.5 | −14.7 | −14.9 |
| London | 1399.2 (272.7) | 1263.1 (236.8) | 870.0 (128.1) | +3.6 | −31.1 | −37.8 |
| South West | 767.9 (64.2) | 805.0 (74.5) | 678.9 (52.0) | +10.0 | −15.7 | −11.6 |
| East of England | 764.0 (48.9) | 801.7 (49.9) | 658.3 (42.1) | +11.1 | −17.9 | −13.8 |
| South East | 754.0 (81.3) | 742.9 (75.1) | 651.6 (61.9) | +4.7 | −12.3 | −13.6 |
| Percentage male | | | | | | |
| <48.7% | 825.5 (169.8) | 851.5 (172.3) | 687.6 (86.2) | +10.0 | −19.3 | −16.7 |
| 48.7%–49.1% | 855.1 (240.0) | 859.5 (192.8) | 658.8 (71.0) | +8.0 | −23.4 | −23.0 |
| 49.1%–49.5% | 811.2 (201.6) | 834.0 (173.9) | 674.7 (85.1) | +9.6 | −19.1 | −16.8 |
| >49.5% | 907.9 (350.1) | 906.6 (262.9) | 725.5 (131.7) | +8.2 | −20.0 | −20.1 |
| Percentage working age | | | | | | |
| <62.6% | 761.2 (60.2) | 786.8 (73.6) | 659.6 (69.7) | +8.4 | −16.2 | −13.4 |
| 62.6%–64.5% | 782.8 (141.0) | 771.7 (111.8) | 687.7 (84.2) | +4.0 | −10.9 | −12.2 |
| 64.5%–66.3% | 786.7 (117.9) | 838.9 (153.9) | 700.7 (93.1) | +13.2 | −14.5 | −10.9 |
| >66.3% | 985.5 (347.5) | 995.5 (275.0) | 812.2 (151.8) | +7.6 | −18.4 | −17.6 |
| GDHI per capita* | | | | | | |
| <£15 892 | 884.8 (198.6) | 956.1 (174.3) | 740.0 (94.0) | +13.7 | −22.6 | −16.4 |
| £15 892–£18 346 | 782.9 (154.5) | 795.8 (94.7) | 659.9 (72.3) | +11.7 | −17.1 | −15.7 |
| £18 346–£21 422 | 824.6 (245.3) | 816.3 (173.0) | 656.8 (67.0) | +6.0 | −19.5 | −20.4 |
| >£21 422 | 893.9 (320.5) | 891.9 (297.1) | 708.5 (123.8) | +7.2 | −20.6 | −20.7 |
| Unemployment rate | | | | | | |
| <3.8% | 722.9 (48.7) | 785.9 (75.8) | 658.2 (70.2) | +19.9 | −16.3 | −9.0 |
| 3.8%–5.1% | 802.5 (117.0) | 732.2 (58.6) | 690.2 (92.8) | −2.1 | −5.7 | −14.0 |
| 5.1%–6.8% | 926.7 (177.4) | 761.0 (83.8) | 801.8 (141.5) | −12.3 | +5.4 | −13.5 |
| >6.8% | 1352.3 (335.9) | 972.6 (227.9) | 842.4 (64.0) | −22.8 | −13.4 | −37.7 |
| IMD† | | | | | | |
| 1 (most deprived) | 1068.2 (326.0) | 1084.2 (242.4) | 782.5 (121.7) | +10.6 | −27.8 | −26.8 |
| 2 | 883.2 (246.4) | 914.2 (201.6) | 713.0 (116.2) | +10.5 | −22.0 | −19.3 |
| 3 | 798.0 (136.5) | 809.0 (97.1) | 664.0 (54.3) | +7.6 | −17.9 | −16.8 |
| 4 | 746.4 (120.1) | 769.5 (102.1) | 647.3 (74.5) | +10.1 | −15.5 | −13.3 |
| 5 (least deprived) | 732.1 (89.6) | 733.1 (82.6) | 648.0 (62.0) | +6.8 | −11.6 | −11.5 |
| Level of reductions to working age benefits per capita | | | | | | |
| 1 (<£457) | 726.6 (84.4) | 731.3 (82.1) | 643.6 (65.9) | +7.4 | −12.0 | −11.4 |
| 2 (£457–£575) | 805.8 (222.8) | 819.3 (186.4) | 675.5 (96.0) | +8.6 | −17.6 | −16.2 |
| 3 (£575–£690) | 868.8 (261.0) | 885.8 (197.0) | 694.2 (107.6) | +9.2 | −21.6 | −20.1 |
| 4 (>£690) | 985.3 (262.5) | 1013.4 (208.1) | 751.7 (108.8) | +11.3 | −25.8 | −23.7 |

*GDHI is gross disposable household income per capita per year, an area-level measure of individuals' available money for spending or saving following payment of taxes and receipt of benefits.
†IMD, quintiles based on relative ranking of LAs.
IMD, Index of Multiple Deprivation; LA, local authority.

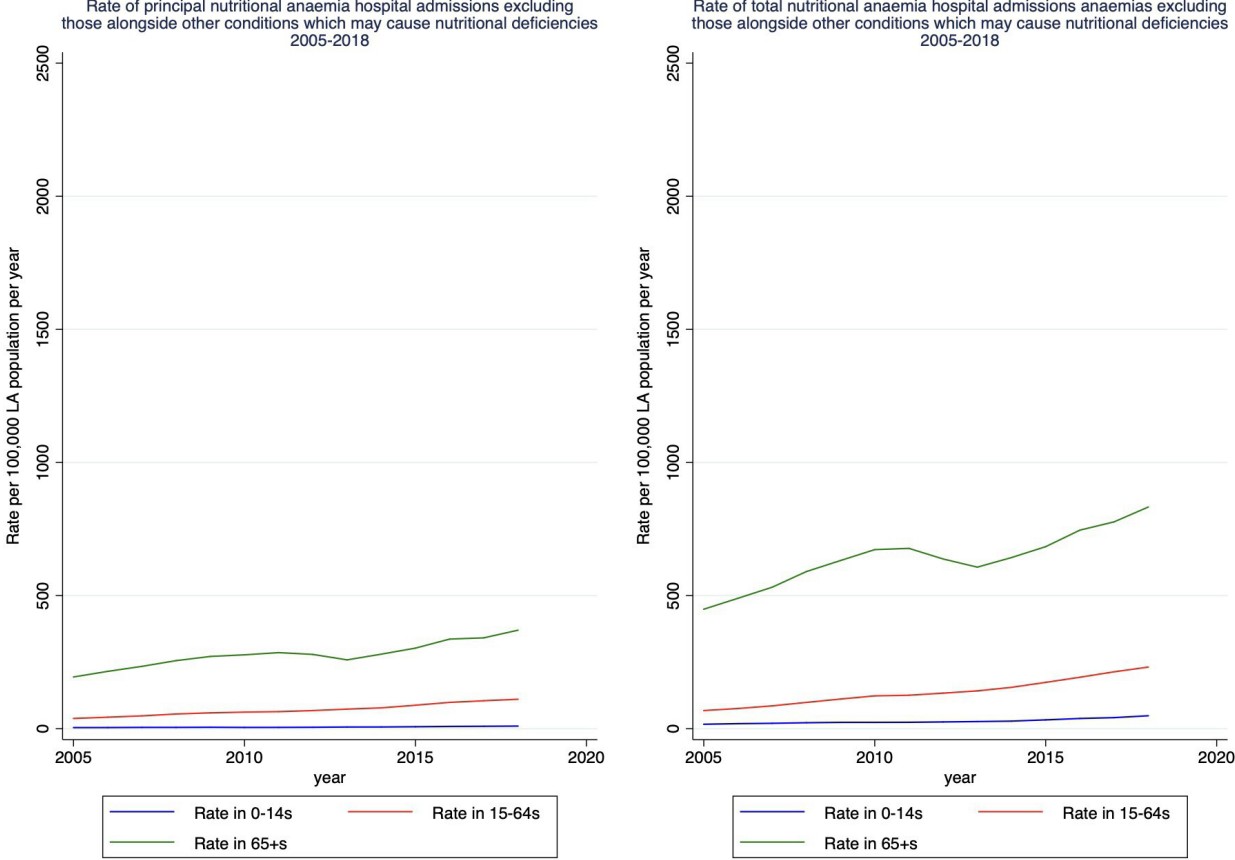

**Figure 2** Rates of principal and total nutritional anaemia hospital admissions by year, stratified by age. LA, local authority.

2013, and then increasing more steeply between 2014 and 2018. Rate of admissions in 65+s was 684 per 100 000 population in 2005 and 2103 per 100 000 in 2018. There was also an increase throughout the study period for 15–64 year olds, from 88 per 100 000 population in 2005 to 364 per 100 000 population. Total nutritional anaemia hospital admissions also increased slightly in 0–14s between 2005 and 2018, from 19 to 57 admissions per 100 000 population. Rates of admissions also increased for both men and women when stratified by sex, and women had a higher rate of nutritional anaemias than men (figure 3).

In 2018, the rate of principal nutritional anaemia hospital admissions was 236 per 100 000 population (table 2). Rates increased with age (11 per 100 000 population for 0–14s, 154 for 15–64s and 717 for 65+s). The rate of total nutritional anaemia hospital admissions in 2018 was 632 per 100 000 population. Rates of total nutritional anaemia hospital admissions also increased with age—rates were relatively low in children (57 per 100 000 population) but were 364 and 2102 per 100 000 population in adults (15–64 and 65+, respectively). Principal and total nutritional anaemia hospital admissions increased with deprivation for all age groups. Rates of admissions increased with unemployment and level of benefit reductions and decreased with GDHI.

### Poisson fixed effects analysis of the association between total LA service spending and principal and total nutritional anaemia hospital admissions

A £100 per capita higher LA service spending was associated with 1.9% lower total nutritional anaemia hospital admissions (adjusted incidence rate ratio (IRR): 0.981, 95% CI: 0.964 to 0.999) (table 3). This relationship was only statistically significant in adults aged ≥15 years. A £100 higher LA service spend per capita was associated with 2.2% lower total nutritional anaemia hospital admissions in the adjusted model for people aged 15–64, and 2.1% lower rates in 65+, respectively (IRR for 15–64: 0.978, 95% CI: 0.959 to 0.998; IRR for 65+: 0.979, 95% CI: 0.960 to 0.997). When stratified by IMD, this relationship was statistically significant in the first (most deprived) (adjusted IRR for all age groups: 0.974, 95% CI: 0.951 to 0.998) and fourth (adjusted IRR for all age groups: 0.883, 95% CI: 0.831 to 0.939) quintiles. This persisted in adults. When stratified by level of reductions to working age benefits, this relationship was present in the second quartile (adjusted IRR for all age groups: 0.963, 95% CI: 0.935 to 0.991) and fourth quartile for 65+ (adjusted IRR for 65+: 0.973, 95% CI: 0.947 to 0.999).

When we investigated this relationship stratified by sex, we found that this relationship was statistically significant for men only (table 4) (online supplemental

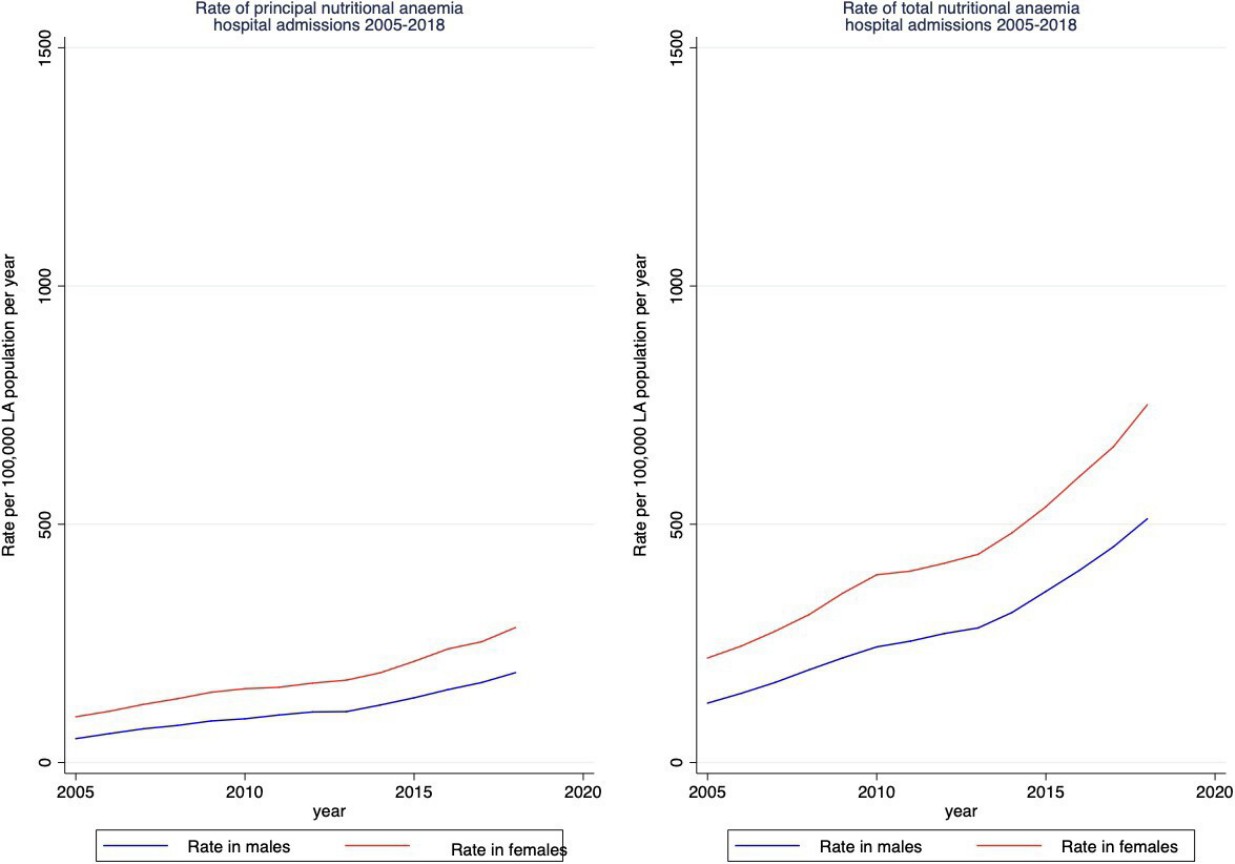

**Figure 3** Rates of principal and total nutritional anaemia hospital admissions by year, stratified by sex. LA, local authority.

appendix 3). A £100 per capita higher LA service spending was associated with 2.5% lower total nutritional anaemia admissions in men (adjusted IRR 0.975, 95% CI: 0.957 to 0.993). Similarly to above, the relationship was statically significant in the first (adjusted IRR: 0.968, 95% CI: 0.944 to 0.993) and fourth (adjusted IRR: 0.872, 95% CI: 0.817 to 0.932) IMD quintiles for men, but none of the results were statistically significant for women when stratified by IMD. When stratified by reductions to working age benefits, higher LA spending was statistically significantly associated with lower hospital admissions for nutritional anaemias in the second (adjusted IRR: 0.955, 95% CI: 0.926 to 0.985) and fourth (adjusted IRR: 0.970, 95% CI: 0.942 to 1.000) quartiles for men only.

When we investigated this relationship with principal rather than total nutritional anaemia hospital admissions as the outcome, this relationship was no longer statistically significant (adjusted IRR: 0.987, 95% CI: 0.963 to 1.011) (online supplemental appendix 4). LA spending was not statistically significantly associated with principal nutritional anaemia hospital admissions in any of the age groups: 0–14 (adjusted IRR: 0.996, 95% CI: 0.945 to 1.051), 15–64 (adjusted IRR: 0.986, 95% CI: 0.957 to 1.015) and 65+ (adjusted IRR: 0.983, 95% CI: 0.957 to 1.009). When stratified by IMD, a statistically significant relationship was present for the fourth quintile only (eg, IRR for all ages: 0.838, 95% CI: 0.759 to 0.925).

### Sensitivity analysis: excluding diagnoses which may cause a nutritional deficiency

The increase in nutritional anaemias over time (figure 2) was much less steep when nutritional anaemia admissions alongside a diagnosis of a condition which may cause a nutritional deficiency were excluded (online supplemental appendix 5). When nutritional anaemia hospital admissions that were alongside a principal or secondary diagnosis of another condition which may cause a nutritional deficiency were excluded in the Poisson regression analysis, generally the relationship between LA spending and total nutritional anaemia hospital admissions was not statistically significant (eg, adjusted IRR for all age groups: 0.991, 95% CI: 0.974 to 1.009) (online supplemental appendix 6). There were some significant relationships in the middle IMD and level of reductions to working age benefits (eg, adjusted IRR for quintile 2 of reductions to working age benefits for 15–64: 0.960, 95% CI: 0.924 to 0.997).

### Sensitivity analysis: total LA service expenditure excluding social care

When social care was excluded from the exposure variable, a £100 increase in LA service spending was associated with a 4.0% decrease in total nutritional anaemia hospital admissions (adjusted IRR: 0.960, 95% CI: 0.939 to 0.982) (online supplemental appendix 7). This relationship remained in the 15–64 age group (adjusted IRR

**Table 2** Mean rate of principal and total nutritional anaemia hospital admissions per 100 000 LA population in 2018, stratified by socio-demographic variables

Mean rate of nutritional anaemia hospital admissions per 100000 LA population in 2018. Mean (SE in brackets)

| | All ages | | 0–14 | | 15–64 | | 65+ | |
|---|---|---|---|---|---|---|---|---|
| | Principal nutritional anaemia hospital admissions | Total nutritional anaemia hospital admissions | Principal nutritional anaemia hospital admissions | Total nutritional anaemia hospital admissions | Principal nutritional anaemia hospital admissions | Total nutritional anaemia hospital admissions | Principal nutritional anaemia hospital admissions | Total nutritional anaemia hospital admissions |
| Total | 236.4 (95.8) | 632.4 (213.7) | 11.1 (16.9) | 56.8 (41.1) | 154.1 (65.0) | 363.6 (145.8) | 717.1 (278.3) | 2101.8 (765.5) |
| Region | | | | | | | | |
| North East | 306.7 (106.1) | 938.4 (268.3) | 13.5 (6.9) | 67.7 (26.2) | 208.6 (79.3) | 553.2 (210.4) | 898.1 (329.1) | 2997.9 (851.1) |
| North West | 284.5 (88.6) | 836.4 (200.5) | 22.4 (35.4) | 83.9 (62.5) | 189.6 (72.8) | 492.8 (172.2) | 854.7 (308.1) | 2688.0 (753.8) |
| Yorkshire and the Humber | 278.3 (73.7) | 698.0 (149.8) | 7.9 (9.9) | 55.2 (36.5) | 181.9 (55.4) | 404.8 (127.4) | 804.8 (214.0) | 2154.9 (374.2) |
| East Midlands | 206.7 (86.1) | 531.5 (179.1) | 6.7 (8.8) | 35.7 (29.0) | 136.7 (61.1) | 299.9 (113.1) | 596.0 (261.4) | 1677.8 (557.2) |
| West Midlands | 211.0 (70.6) | 584.5 (155.0) | 11.1 (10.7) | 61.9 (42.6) | 146.5 (52.8) | 347.9 (113.2) | 585.8 (199.5) | 1788.7 (503.3) |
| London | 183.2 (62.5) | 644.2 (140.2) | 13.7 (9.1) | 85.4 (29.3) | 137.5 (49.4) | 415.1 (109.8) | 719.9 (209.0) | 2956.2 (763.2) |
| South West | 271.2 (119.0) | 628.3 (204.4) | 6.8 (8.4) | 43.2 (28.7) | 164.0 (69.0) | 332.6 (108.1) | 730.1 (313.1) | 1825.0 (592.2) |
| East of England | 255.2 (115.1) | 620.2 (217.7) | 9.8 (14.3) | 47.7 (32.1) | 157.2 (72.1) | 354.6 (127.1) | 786.0 (304.0) | 1985.4 (582.0) |
| South East | 205.8 (71.6) | 515.4 (150.5) | 9.5 (14.2) | 48.1 (34.5) | 127.7 (48.1) | 276.6 (90.3) | 651.3 (232.8) | 1758.6 (571.3) |
| Percentage male | | | | | | | | |
| <48.7% | 238.0 (99.8) | 614.5 (216.4) | 9.7 (13.4) | 48.9 (32.3) | 144.9 (60.9) | 332.2 (127.0) | 662.1 (249.0) | 1834.3 (581.6) |
| 48.7%–49.1% | 250.4 (100.3) | 646.5 (206.5) | 8.1 (9.6) | 47.3 (31.1) | 153.4 (61.9) | 351.9 (128.1) | 725.0 (298.2) | 1981.2 (625.1) |
| 49.1%–49.5% | 240.4 (104.9) | 633.1 (248.4) | 12.3 (16.5) | 57.9 (43.3) | 159.7 (76.5) | 364.8 (170.2) | 725.3 (305.2) | 2082.4 (844.3) |
| >49.5% | 223.8 (81.3) | 635.5 (186.0) | 12.9 (22.0) | 66.9 (47.5) | 156.0 (59.4) | 390.7 (142.7) | 742.7 (260.1) | 2372.5 (805.6) |
| Percentage working age | | | | | | | | |
| <62.6% | 241.5 (103.4) | 624.5 (223.7) | 9.2 (14.3) | 47.8 (37.5) | 148.4 (65.3) | 337.0 (139.1) | 662.6 (275.8) | 1826.9 (616.5) |
| 62.6%–64.5% | 256.9 (85.8) | 681.2 (223.6) | 12.0 (11.2) | 60.9 (38.5) | 174.3 (67.9) | 411.4 (167.4) | 801.5 (259.2) | 2278.9 (690.9) |
| 64.5%–66.3% | 223.0 (85.1) | 591.4 (175.7) | 17.8 (37.8) | 75.1 (59.1) | 163.7 (66.6) | 381.9 (137.9) | 763.8 (298.2) | 2230.1 (704.6) |
| >66.3% | 191.5 (71.8) | 609.0 (168.0) | 12.6 (9.5) | 72.8 (35.2) | 136.1 (49.1) | 375.1 (115.6) | 757.2 (266.2) | 2799.7 (907.6) |
| GDHI* | | | | | | | | |
| <£15892 | 301.8 (102.0) | 817.8 (244.4) | 19.4 (30.4) | 86.8 (59.3) | 208.7 (69.3) | 507.4 (180.5) | 889.2 (281.3) | 2628.6 (746.9) |
| £15 892–£18 346 | 241.0 (103.8) | 649.8 (221.2) | 10.1 (9.4) | 52.1 (31.5) | 156.2 (67.1) | 368.4 (135.5) | 704.1 (288.3) | 2066.1 (692.3) |
| £18 346–£21 422 | 235.7 (90.7) | 598.8 (172.8) | 9.3 (12.9) | 45.7 (30.2) | 151.3 (57.2) | 331.2 (103.6) | 692.8 (295.8) | 1911.2 (630.6) |
| >£21 422 | 200.6 (70.5) | 549.8 (155.5) | 9.1 (12.6) | 53.2 (36.1) | 126.9 (48.3) | 310.9 (109.1) | 656.3 (219.9) | 1995.6 (804.9) |
| Unemployment rate | | | | | | | | |
| <3.8% | 230.6 (96.5) | 579.0 (191.7) | 8.3 (13.1) | 44.1 (31.3) | 140.3 (61.3) | 310.7 (111.1) | 673.5 (278.9) | 1793.7 (579.8) |
| 3.8%–5.1% | 242.8 (92.5) | 656.6 (203.7) | 12.1 (21.6) | 63.4 (47.2) | 164.3 (60.5) | 387.9 (131.7) | 750.3 (276.9) | 2226.3 (670.2) |
| 5.1%–6.8% | 237.5 (102.4) | 752.4 (242.7) | 18.9 (16.1) | 84.0 (39.0) | 175.0 (76.7) | 482.5 (179.4) | 786.8 (256.0) | 2911.8 (887.9) |
| >6.8% | 279.4 (84.1) | 856.6 (256.7) | 19.7 (8.8) | 117.9 (28.0) | 221.2 (60.6) | 602.4 (184.1) | 867.9 (297.3) | 2873.1 (706.2) |

Continued

**Table 2** Continued

| | All ages | | 0–14 | | 15–64 | | 65+ | |
|---|---|---|---|---|---|---|---|---|
| | Mean rate of nutritional anaemia hospital admissions per 100000 LA population in 2018 Mean (SE in brackets) | | | | | | | |
| | Principal nutritional anaemia hospital admissions | Total nutritional anaemia hospital admissions | Principal nutritional anaemia hospital admissions | Total nutritional anaemia hospital admissions | Principal nutritional anaemia hospital admissions | Total nutritional anaemia hospital admissions | Principal nutritional anaemia hospital admissions | Total nutritional anaemia hospital admissions |
| **IMD†** | | | | | | | | |
| 1 (most deprived) | 254.9 (99.1) | 764.9 (230.5) | 20.2 (28.0) | 90.5 (51.6) | 186.5 (75.1) | 493.9 (173.0) | 817.3 (267.4) | 2781.7 (775.2) |
| 2 | 246.6 (113.1) | 670.4 (234.7) | 11.1 (9.3) | 61.4 (36.6) | 165.0 (66.7) | 390.3 (138.7) | 742.9 (321.4) | 2252.2 (847.3) |
| 3 | 241.1 (89.3) | 622.9 (177.6) | 11.0 (13.5) | 51.9 (32.2) | 156.2 (57.2) | 350.2 (103.7) | 710.8 (275.3) | 1993.3 (602.7) |
| 4 | 238.8 (94.4) | 595.1 (180.2) | 7.9 (13.6) | 41.9 (30.8) | 142.8 (59.6) | 314.4 (106.7) | 692.2 (277.3) | 1829.7 (541.6) |
| 5 (least deprived) | 202.3 (73.3) | 512.3 (153.8) | 5.5 (8.9) | 38.5 (28.1) | 120.9 (45.6) | 271.2 (82.4) | 625.1 (214.7) | 1665.2 (467.5) |
| Level of reductions to working age benefits per capita | | | | | | | | |
| 1 (<£457) | 206.2 (75.7) | 521.9 (150.1) | 5.3 (8.5) | 39.4 (30.0) | 123.0 (46.8) | 275.7 (82.5) | 630.4 (231.7) | 1679.5 (484.8) |
| 2 (£457–£575) | 238.8 (100.4) | 610.6 (196.9) | 11.5 (16.1) | 50.2 (33.5) | 145.3 (60.7) | 327.4 (111.3) | 703.0 (284.4) | 1962.2 (717.5) |
| 3 (£575–£690) | 239.5 (105.0) | 639.8 (201.6) | 9.3 (8.6) | 51.6 (30.9) | 158.2 (62.7) | 366.4 (115.2) | 744.6 (305.4) | 2208.3 (748.8) |
| 4 (>£690) | 263.3 (92.7) | 759.8 (233.6) | 18.9 (26.3) | 88.0 (51.4) | 192.4 (70.0) | 489.5 (172.6) | 795.5 (263.7) | 2571.1 (801.0) |

*GDHI is gross disposable household income per capita per year, an area-level measure of individuals' available money for spending or saving following payment of taxes and receipt of benefits.
†IMD, quintiles based on relative ranking of LAs.
IMD, Index of Multiple Deprivation; LA, local authority.

**Table 3** Association between LA service spending and total nutritional anaemia hospital admissions, shown as the incident rate ratio and stratified by age (95% CIs in brackets)

| | Incidence rate ratio for total nutritional anaemia hospital admissions by LA with a £100 increase in total LA service spending | | | |
| --- | --- | --- | --- | --- |
| | All ages | 0–14 | 15–64 | 65+ |
| Full sample* | 0.981 (0.964 to 0.999) p=0.038 | 0.974 (0.904 to 1.050) p=0.488 | 0.978 (0.959 to 0.998) p=0.033 | 0.979 (0.960 to 0.997) p=0.024 |
| IMD† | | | | |
| 1 (most deprived) | 0.974 (0.951 to 0.998) p=0.032 | 0.943 (0.834 to 1.066) p=0.348 | 0.971 (0.943 to 0.999) p=0.043 | 0.974 (0.950 to 0.999) p=0.038 |
| 2 | 0.991 (0.949 to 1.035) p=0.678 | 1.030 (0.980 to 1.082) p=0.251 | 0.999 (0.957 to 1.043) p=0.960 | 0.982 (0.936 to 1.030) p=0.452 |
| 3 | 1.016 (0.976 to 1.058) p=0.431 | 1.032 (0.938 to 1.135) p=0.524 | 1.019 (0.968 to 1.073) p=0.472 | 1.006 (0.966 to 1.047) p=0.779 |
| 4 | 0.883 (0.831 to 0.939) p<0.001 | 0.929 (0.790 to 1.092) p=0.371 | 0.851 (0.792 to 0.915) p<0.001 | 0.891 (0.838 to 0.946) p<0.001 |
| 5 (least deprived) | 1.011 (0.955 to 1.070) p=0.712 | 1.014 (0.860 to 1.195) p=0.870 | 0.989 (0.922 to 1.063) p=0.768 | 1.018 (0.966 to 1.072) p=0.505 |
| Level of reductions to working age benefits per capita | | | | |
| 1 (lowest reductions) | 1.011 (0.961 to 1.063) p=0.680 | 1.013 (0.877 to 1.170) p=0.864 | 1.000 (0.946 to 1.056) p=0.985 | 1.013 (0.962 to 1.067) p=0.616 |
| 2 | 0.963 (0.935 to 0.991) p=0.010 | 0.955 (0.889 to 1.026) p=0.208 | 0.954 (0.920 to 0.989) p=0.010 | 0.963 (0.935 to 0.992) p=0.012 |
| 3 | 0.992 (0.952 to 1.032) p=0.671 | 1.029 (0.928 to 1.141) p=0.587 | 0.995 (0.955 to 1.036) p=0.803 | 0.985 (0.942 to 1.030) p=0.500 |
| 4 (greatest reductions) | 0.975 (0.949 to 1.003) p=0.077 | 0.935 (0.824 to 1.061) p=0.294 | 0.973 (0.941 to 1.006) p=0.104 | 0.973 (0.947 to 0.999) p=0.043 |

*Adjusted by percentage working age, percentage male, GDHI and unemployment rate.
†IMD, quintiles based on relative ranking of LAs.
GDHI, gross disposable household income ; IMD, Index of Multiple Deprivation ; LA, local authority.

for 15–64: 0.958, 95% CI: 0.934 to 0.983) and the 65+ age group (adjusted IRR for 65+: 0.958, 95% CI: 0.936 to 0.980) when stratified by age. Statistically significantly greater reductions in nutritional anaemias were seen in the first and fourth IMD quintiles, suggesting effect modification by deprivation. Little effect modification was seen by level of reductions to working age benefits.

## DISCUSSION

This analysis of data from 2005 to 2018 found that reductions in spending due to austerity policies may have increased nutritional anaemia hospital admissions, especially in more deprived areas. LA service spending increased by 9% between 2005 and 2009 then decreased by 20% between 2010 and 2018, following the introduction of austerity policies in 2010. Total nutritional anaemia hospital admissions also increased between 2005 and 2018. We found that a £100 higher LA service spending was associated with 1.9% lower total nutritional anaemia hospital admissions. This was driven by admissions in adults. A 2.6% lower admission rate was seen with a £100 higher expenditure in the most deprived LAs, suggesting greater impacts by deprivation, especially since these LAs

experienced greater service spending reductions overall (a 28% reduction was seen in the most deprived quintile compared with a 12% reduction in the least deprived quintile between 2010 and 2018). However, we also found that a £100 higher expenditure was statistically significantly associated with a reduction in admissions in quintile 4 (one of the least deprived quintiles), suggesting that there may be impacts across the spectrum of LAs in terms of deprivation. Regardless of the relative effects of spending across levels of deprivation, the far greater cuts to spending in more deprived areas mean the absolute impact on nutritional anaemias will have been felt disproportionately in these areas.

We found that total nutritional anaemia hospital admissions increased between 2005 and 2018. This aligns with previous research suggesting decreases in micronutrient consumption and corresponding increases in nutritional anaemias in the UK.[5 55 56] We identified an overall increase in total nutritional anaemia hospital admissions which plateaued between 2010 and 2013, and then increased again. Research has found that between 2007 and 2011 the percentage of patients known to have had nutritional screening increased from 67–78% to

**Table 4** Impact of LA service spending on total nutritional anaemia hospital admissions, shown as the incident rate ratio and stratified by sex (95% CIs in brackets)

| | Incidence rate ratio for total nutritional anaemia hospital admissions by LA with a £100 increase in total LA service spending | | |
| --- | --- | --- | --- |
| | **All ages** | **Male** | **Female** |
| Full sample* | 0.981 (0.964 to 0.999) p=0.038 | 0.975 (0.957 to 0.993) p=0.007 | 0.992 (0.974 to 1.010) p=0.358 |
| IMD† | | | |
| 1 (most deprived) | 0.974 (0.951 to 0.998) p=0.032 | 0.968 (0.944 to 0.993) p=0.012 | 0.980 (0.956 to 1.005) p=0.118 |
| 2 | 0.991 (0.949 to 1.035) p=0.678 | 0.988 (0.947 to 1.030) p=0.563 | 1.005 (0.962 to 1.050) p=0.818 |
| 3 | 1.016 (0.976 to 1.058) p=0.431 | 1.009 (0.962 to 1.058) p=0.707 | 1.032 (0.989 to 1.076) p=0.151 |
| 4 | 0.883 (0.831 to 0.939) p<0.001 | 0.872 (0.817 to 0.932) p<0.001 | 0.902 (0.851 to 0.956) p<0.001 |
| 5 (least deprived) | 1.011 (0.955 to 1.070) p=0.712 | 1.001 (0.936 to 1.071) p=0.967 | 1.029 (0.967 to 1.094) p=0.367 |
| Level of reductions to working age benefits per capita | | | |
| 1 (lowest reductions) | 1.011 (0.961 to 1.063) p=0.680 | 1.013 (0.962 to 1.067) p=0.623 | 1.020 (0.962 to 1.083) p=0.506 |
| 2 | 0.963 (0.935 to 0.991) p=0.010 | 0.955 (0.926 to 0.985) p=0.004 | 0.981 (0.951 to 1.011) p=0.210 |
| 3 | 0.992 (0.952 to 1.032) p=0.671 | 0.983 (0.944 to 1.024) p=0.419 | 0.999 (0.960 to 1.040) p=0.960 |
| 4 (greatest reductions) | 0.975 (0.949 to 1.003) p=0.077 | 0.970 (0.942 to 1.000) p=0.048 | 0.986 (0.957 to 1.015) p=0.330 |

*Adjusted by percentage working age, percentage male, GDHI and unemployment rate.
†IMD, quintiles based on relative ranking of LAs.
GDHI, gross disposable household income ; IMD, Index of Multiple Deprivation ; LA, local authority.

86–95%.[57] Therefore, the increase in nutritional anaemia hospital admissions between 2005 and 2010 may be an artefact of increased screening over this period, with the plateau over the following years representing nutritional screening reaching near-saturation of hospital admissions. Given that nutritional screening was approaching saturation as austerity was being rolled out, an increase in nutritional anaemias after 2010—during austerity—may be in part explained by changes in LA spending as per our hypothesis.

We found that the impact of LA spending on nutritional anaemia admissions was statistically significant in men but not women. This may be due to biological mechanisms—men have higher dietary reference values than women for most micronutrients (with the exception of iron to cover for iron losses during menstruation for women), although there is a lack of nuance in these reference values with regards to more subtle differences in growth and maturation rates and health vulnerabilities.[58] Analyses of UK National Diet and Nutrition Survey data have found that women were more vulnerable than men to micronutrient deficiencies. It is possible that some women vulnerable to micronutrient deficiencies were already likely to be admitted with nutritional anaemias prior to austerity policies, particularly due to monitoring during pregnancy, whereas these impacts may have pushed men who were vulnerable to deficiencies into being admitted with nutritional anaemias. With regards to differences in causal pathways, it has been suggested that women have been more affected by the UK government's austerity policies than men.[59] Thus, we might have expected to see greater or more statistically significant effects in women. Further

individual level research, taking into account causal pathways, is needed to understand these sex differences.

Our study suggests that these decreases in LA service spending may have health impacts. This supports other studies which suggest that decreases in LA service spending may adversely affect health and mortality.[14 60] Our research adds new evidence that there may be associations with nutritional anaemias in addition to other health outcomes previously examined. As nutritional anaemias are important risk factors for diseases, this may be one pathway through which reductions in LA service expenditure may be associated with health. More deprived areas also experienced greater reductions in LA service spending, meaning the absolute increase in nutritional deficiencies attributable to service spending cuts will have been greater in more deprived areas, which already have greater health needs. This aligns with other evidence that austerity measures had a greater impact in more deprived areas and may have widened health inequalities due to the role of LA services in improving the social determinants of health.[9 10 12 13]

This research has several strengths. We are the first to examine associations between austerity policies and nutritional anaemia hospital admissions. We used a large panel data set with coverage of all NHS hospitals for 312 LAs in England, over a long time period (2005–2018). We used a fixed effects modelling strategy and were able to account for all time-invariant factors that differ between LAs and changing population sizes, allowing us to assess within-LA change in hospital admissions over time. Furthermore, our sensitivity analysis found a stronger relationship between total LA service spending and total nutritional

anaemia hospital admissions when social care spending was excluded. This suggests that social care spending may be blunting the relationship between LA service spending and total nutritional anaemia hospital admissions, likely due to higher levels of nutritional anaemias requiring higher levels of social care spending.[54] This further supports our hypothesis that austerity measures may have led to increased nutritional anaemia hospital admissions.

Our study also has some limitations. As previously described, we used counts of nutritional anaemia hospital admissions, which may be an artefact of hospital processes such as more screening leading to more diagnoses. As we used data on hospital admissions, we did not examine diagnoses bygeneral practice, which may give a more complete picture of potential impacts on nutritional anaemias at the population level. We were also unable to break admissions down by severity, or to access data from laboratory tests, which may have picked up more cases of anaemia than were coded in the HES data. Further research could usefully explore these issues, as well as potential impacts of anaemia on the need for transfusions. Furthermore, the relationship we have described was only statistically significant when total nutritional anaemia hospital admissions were included as the outcome measure and not when principal nutritional anaemia hospital admissions were the outcome. Therefore, it is possible that the relationship seen may be a result of LA service spending being associated with increases in people accessing hospitals for other health conditions and nutritional anaemias also being present, leading to more total nutritional anaemia hospital admissions which would otherwise have been in the community and undiagnosed. One possibility is that such increases in hospital access may be due to more general impacts of austerity policies on health outcomes. Additionally, our exposure data were per financial year and our outcome data were per calendar year, representing a potential mismatch in time periods assessed. Another limitation is that we did not obtain outcome data by both age and sex.

Sensitivity analyses excluding hospital admissions alongside a medical condition which may cause a nutritional deficiency did not identify a relationship between LA service spending and total nutritional anaemia hospital admissions. This suggests that the relationship between LA spending and nutritional anaemia hospital admissions is driven by changes in the prevalence of these other conditions. The degree to which nutritional anaemias may be caused by other medical conditions is difficult to assess in routine data without more detailed clinical parameters and we excluded a range of common conditions such as cancers and type 2 diabetes where nutritional deficiencies are common. These diseases have a nutritional component, particularly with regards to their association with obesity, and thus may represent wide-ranging impacts of austerity policies on nutrition in terms of both nutritional anaemias and chronic disease. Together our analyses suggest that the burden of nutritional anaemias is increasing over time and that this is

more apparent in areas with greater levels of deprivation, greater levels of medical comorbidity and potentially in areas with more severe reductions in LA spending.[61]

Our panel study suggests that reductions in LA service spending due to UK austerity policies may be associated with an increase in nutritional anaemias. This adds further evidence to the growing evidence base regarding potential impacts of these policies on health and health inequalities. Nutritional anaemias have wide-ranging impacts on health. Thus, policymakers should consider associations between fiscal policies and health prior to their introduction. This is particularly pertinent given the budget deficit as well as possible further reductions in LA budgets alongside tax rises coupled with an emerging cost of living crisis. In the context of the proposed levelling-up agenda in England, this study adds more evidence to support the need for reinvestment in LA services as a means to improve population health outcomes and reduce health inequalities.[62] Nutritional anaemias fell between 2005 and 2009, a period of a national health inequalities strategy in England which entailed sustained investment in LAs, the NHS, prevention and the welfare system (including tax credits, supplemental funds for pregnant women and increases in the national pension).[63] Policymakers should draw on this past success and consider including health outcomes in the allocation criteria for levelling-up investments.[64] Indeed, a national health inequalities strategy is once more needed to level up health across the country. However, individual-level research is needed to elucidate pathways between changes to LA service spending and nutritional anaemias, particularly with regards to different areas of LA service spending.

## Conclusion

Overall our analyses conclude that increased LA spending was associated with reduced hospital admissions for nutritional anaemia, with austerity having the opposite effect. This research adds to other accumulating evidence of the negative impacts of austerity and the importance of maintaining and increasing public sector spending.

**Acknowledgements** We are grateful to Christina Beatty and Steve Fothergill for sharing their data from The Uneven Impact of Welfare Reform project.

**Contributors** Study conception and design by RJ, EPV, KEM and AAL. Data analysis was by RJ with input from RJ, EPV, KEM and AAL. RJ wrote the first draft and all other authors (EPV, KEM, KD, DT-R, CB, CM and AAL) made significant contributions to the final version. RJ is guarantor

**Funding** This study is funded by the National Institute for Health Research (NIHR) School for Public Health Research (SPHR), Grant Reference Number PD-SPH-2015. DT-R is funded by the MRC on a Clinician Scientist Fellowship (MR/P008577/1). The views expressed are those of the author(s) and not necessarily those of the NIHR or the Department of Health and Social Care.

**Competing interests** None declared.

**Patient and public involvement** Patients and/or the public were involved in the design, or conduct, or reporting, or dissemination plans of this research. Refer to the Methods section for further details.

**Patient consent for publication** Not applicable.

**Ethics approval** Not applicable.

**Provenance and peer review** Not commissioned; externally peer reviewed.

**Data availability statement** Data are available in a public, open access repository. All data are available on the PLDR resource (pldr.org).

**ORCID iDs**
Rosemary Jenkins http://orcid.org/0000-0003-4600-2258
Kate E Mason http://orcid.org/0000-0001-5020-5256
Konstantinos Daras http://orcid.org/0000-0002-4573-4628
Anthony A Laverty http://orcid.org/0000-0003-1318-8439

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
