## [Reviewer comments · BMJ Open]

ARTICLE DETAILS

TITLE (PROVISIONAL)	Local area public sector spending and nutritional anaemia hospital admissions in England: a longitudinal ecological study
AUTHORS	Jenkins, Rosemary; Vamos, Eszter; Mason, Kate; Daras, Konstantinos; Taylor-Robinson, David; Bambra, Clare; Millett, Christopher; Laverty, Anthony

VERSION 1 – REVIEW

REVIEWER	James, Beki Leeds Children's Hospital
REVIEW RETURNED	09-Jan-2022

GENERAL COMMENTS	It was a pleasure to read this important and impressive piece of work, which is written with clarity and fluency. My only comments are relatively minor and do not impact my strong recommendation that the paper be accepted in full. My comments, listed in the order they occurred to me as I read the paper, are as follows: 1. in future work it might be helpful to include patients as key stakeholders, especially from the more deprived LA, as they are true experts. In particular, they might be able to comment on additional factors contributing to the development of NA of which you might be unaware.2. I think that it is critical to include admissions where NA was the secondary reason - as you do - for two reasons: firstly, where there are competing reasons for admission the decision to list one first is not always made by a clinician and so cannot be assumed to be the more important (and in this group there are likely to be co-morbidities so including only these where listed first will miss cases); and, secondly, as the multisystem impact of NA means that the primary listed cause may in some cases be considered to be a manifestation of NA as it is the anaemia which will have "tipped the patient over the edge" and so brought to light the diagnosis of NA.3. I would have liked an overarching sentence at the start of the discussion which summed the findings up and gave the context.4. In the discussion, there is reference to "after" the pandemic. It may be a while until we emerge from the pandemic and so I think that the next step is to look at what is happening during the pandemic (for example, see http://dx.doi.org/10.1136/archdischild-2021-322217).5. Similarly, in future work it would be helpful to look at the effect of the NA on other outcomes such as need for transfusion and death (https://www.judiciary.uk/publications/maya-zab-prevention-of-future-deaths-report/)6. In future work it would also be helpful to compare the completeness of HES data ascertainment with cases identified by
---

	identifying those directly from laboratory data - for example identifying all cases with anaemia, low ferritin/iron/B12 or folate. In our work we found that not all cases identified through laboratory data had been coded as NA. 6. This is clearly a grave situation for all affected, but there is a particular need to understand and act in cases where the individual is vulnerable, for example by virtue of age, where there are safeguarding implications. Thank you again for producing such an excellent and compelling piece of work which has national implications.
--	--

REVIEWER	Sanabria, Marta Cristina Universidad Nacional de Asunción
REVIEW RETURNED	18-Jan-2022

GENERAL COMMENTS	This is a very interesting article competing public health. My suggestion would be to include which criteria for anemia classification was used in the study (WHO?) and which was the haemoglobin cut-off level used I would also like if the authors could specify the percentage of mild, moderate and severe anemia found in the study, iron deficiency anemia. Are there programs in England in charge of producing iron fortified foods?
--

REVIEWER	Alexiou, Alexandros University of Liverpool Faculty of Health and Life Sciences, Department of Public Health and Policy
REVIEW RETURNED	08-Apr-2022

GENERAL COMMENTS	Thanks for inviting me to review this manuscript. I think the study addresses important issues and adds to the evidence base on the topic of the health impacts of austerity. It is well written, data and methods are clearly described, and results discussed in context. I see that the authors have put a lot of effort in this study, and I have no major suggestions for improvements, but there are some issues that I find need attention. 1. I think the study would benefit with a more comprehensive logic model regarding the pathways of local authority (LA) service spending and nutritional anaemia. I think Figure 1 does a poor job in describing associations. For instance, I would argue there are direct (e.g. social care, regulatory services) and indirect pathways (e.g. through disposable income / food purchasing). Some associations stemming from transport services seem rather weak, and don't seem to be supported by the literature referenced. Cultural services are oddly missing; are there potential pathways between cultural services, including access to green space, and changes in diets through e.g. mental health and/or stress? I suggest broadening the model to make it more comprehensive. I would also suggest expanding the relevant section discussing these pathways, which I found rather limited given that it provides the rationale for the study. For instance, housing services are provided as a mechanism example in the abstract, but in the main text this mechanism is only mentioned in passing. 2. Regarding the results and discussion, you mention that results were only statistically significant for males and not females. The
---

	results in Appendix 3 are very interesting, but you barely discuss the implications. Why do you think model results by sex vary? Is this due to potential mechanisms that impact associations? I assume there are biological differences involved but it is difficult to tell, given a) I have little knowledge on the subject and b) there is very little information given regarding the analysis by sex– perhaps another figure, same as figure 2, giving the trends by sex could be useful. I suggest the results by sex should also be highlighted in the text, and the implications discussed in context. Also, it is not clear to me why you decided to focus more on the results stratified by age and not by sex, please elaborate on your decision. From what I understand you don't have data both by age and sex, which is another limitation that should be mentioned. Other minor comments.  - In the data section, can you please clearly state the time period of analysis, and also note that LA annual figures are based on financial years. - Please use "sex" instead of "gender" when you mention stratified data for males and females. - Unfortunately, figure 3 is barely readable so I wasn't able to assess it properly.
--	--

VERSION 1 – AUTHOR RESPONSE

Reviewer 1

It was a pleasure to read this important and impressive piece of work, which is written with clarity and fluency. My only comments are relatively minor and do not impact my strong recommendation that the paper be accepted in full.

We are pleased that the reviewer sees this paper as impressive and that they recommend acceptance. Thank you

1. In future work it might be helpful to include patients as key stakeholders, especially from the more deprived LA, as they are true experts. In particular, they might be able to comment on additional factors contributing to the development of NA of which you might be unaware.

Thank you for this valuable comment – we will take this on board in future research. We did have a Patient and Public Involvement and Engagement panel to which this research was presented, but we agree that we could have involved people with specific expertise in this area and will endeavour to do so in future work.

2. I think that it is critical to include admissions where NA was the secondary reason - as you do - for two reasons: firstly, where there are competing reasons for admission the decision to list one first is not always made by a clinician and so cannot be assumed to be the more important (and in this group there are likely to be co-morbidities so including only these where listed first will miss cases); and, secondly, as the multisystem impact of NA means that the primary listed cause may in some cases be considered to be a manifestation of NA as it is the anaemia which will have "tipped the patient over the edge" and so brought to light the diagnosis of NA.

Thank you for these points which we agree with. We have added additional detail to why we focused on total nutritional anaemia hospital admissions on at the bottom of page 5 and the start of page 6.

3. I would have liked an overarching sentence at the start of the discussion which summed the findings up and gave the context.

Thank you for this comment, we have amended the first sentence of the Discussion to now read “This analysis of data from 2005-18 found that reductions in spending due to austerity policies may have increased nutritional anaemia hospital admissions, especially in more deprived areas”

4. In the discussion, there is reference to "after" the pandemic. It may be a while until we emerge from the pandemic and so I think that the next step is to look at what is happening during the pandemic (for example, see <https://eur01.safelinks.protection.outlook.com/?url=http%3A%2F%2Fdx.doi.org%2F10.1136%2Farchdischild-2021-322217&data=05%7C01%7CRosemary.Jenkins%40birmingham.gov.uk%7Cbf3d74d5688e418da85608da3d9bc1fd%7C699ace67d2e44bcdb303d2bbe2b9bbf1%7C0%7C0%7C637890036429746717%7CUnknown%7CTWFpbGZsb3d8eyJWIjoiMC4wLjAwMDAiLCJQIjoiV2luMzliLCJBTiI6Ikl1haWwiLCJXVCi6Mn0%3D%7C3000%7C%7C%7C&sdata=TsbNZXNCWI9tK36lwQPIUD4pz4aWEzm6XF70AxNScMw%3D&reserved=0>).

Thank you for this point which is of course correct. We have updated this sentence to refer more clearly to the issues of budget deficits and the current cost of living crisis and not refer to ‘after the pandemic’ (page 13, paragraph 3).

5. Similarly, in future work it would be helpful to look at the effect of the NA on other outcomes such as need for transfusion and death (<https://eur01.safelinks.protection.outlook.com/?url=https%3A%2F%2Fwww.judiciary.uk%2Fpublications%2Fmaya-zab-prevention-of-future-deaths-report%2F&data=05%7C01%7CRosemary.Jenkins%40birmingham.gov.uk%7Cbf3d74d5688e418da85608da3d9bc1fd%7C699ace67d2e44bcdb303d2bbe2b9bbf1%7C0%7C0%7C637890036429746717%7CUnknown%7CTWFpbGZsb3d8eyJWIjoiMC4wLjAwMDAiLCJQIjoiV2luMzliLCJBTiI6Ikl1haWwiLCJXVCi6Mn0%3D%7C3000%7C%7C%7C&sdata=hReSfXJ72SW8j3zIDUNqNiirCd4u9qMv95GTu4GR9xo%3D&reserved=0>).

Thank you for this suggestion – it is something we will consider for future work and we now mention this as a potential area for future research on page 13.

6. In future work it would also be helpful to compare the completeness of HES data ascertainment with cases identified by identifying those directly from laboratory data - for example identifying all cases with anaemia, low ferritin/iron/B12 or folate. In our work we found that not all cases identified through laboratory data had been coded as NA.

Thank you for this comment and pointing to other work on the accuracy of coding. We agree that this would be interesting and mention this as an issue on page 13 as well as a possible focus of future research.

7. This is clearly a grave situation for all affected, but there is a particular need to understand and act in cases where the individual is vulnerable, for example by virtue of age, where there are safeguarding implications.

Thank you, we agree with this statement.

Thank you again for producing such an excellent and compelling piece of work which has national implications.

Thank you

Reviewer 2

This is a very interesting article competing public health. My suggestion would be to include which criteria for anemia classification was used in the study (WHO?) and which was the haemoglobin cut-off level used I would also like if the authors could specify the percentage of mild, moderate and severe anemia found in the study, iron deficiency anemia.

Thank you for this comment and your positive review of our article. The Hospital Episode Statistics dataset is a clinical dataset, and thus the data represent clinical diagnoses but does not include detail on cut offs used or similar. We use the ICD-10 codes to determine cases but more granular detail is unfortunately not available in this data. This means that unfortunately we cannot act on this useful suggestion to give a breakdown of the severity of cases. The strength of the HES data however, relies on the fact that it can give a national picture of these issues, albeit without some of the clinical or lab details which would be available in more local datasets. We now mention this more clearly in the limitations section on pages 12 and 13.

Are there programs in England in charge of producing iron fortified foods?

We thank the reviewer for this interesting question. Since the 1940s there has been mandatory fortification of white flour with iron and other micronutrients in the UK. Breakfast cereals are also commonly fortified.

Reviewer 3

Thanks for inviting me to review this manuscript. I think the study addresses important issues and adds to the evidence base on the topic of the health impacts of austerity. It is well written, data and methods are clearly described, and results discussed in context. I see that the authors have put a lot of effort in this study, and I have no major suggestions for improvements, but there are some issues that I find need attention.

We thank the reviewer for this positive assessment of the study.

1. I think the study would benefit with a more comprehensive logic model regarding the pathways of local authority (LA) service spending and nutritional anaemia. I think Figure 1 does a poor job in describing associations. For instance, I would argue there are direct (e.g. social care, regulatory services) and indirect pathways (e.g. through disposable income / food purchasing). Some associations stemming from transport services seem rather weak, and don't seem to be supported by the literature referenced. Cultural services are oddly missing; are there potential pathways between cultural services, including access to green space, and changes in diets through e.g. mental health and/or stress?

I suggest broadening the model to make it more comprehensive. I would also suggest expanding the relevant section discussing these pathways, which I found rather limited given that it provides the rationale for the study. For instance, housing services are provided as a mechanism example in the abstract, but in the main text this mechanism is only mentioned in passing.

We thank the reviewer for this point on our model in Figure 1. We have now updated the logic model to make direct and indirect pathways clearer and added in cultural services following the

reviewer's suggestions (figure 1). We have also expanded the paragraph which discusses the rationale and pathways to give a more comprehensive discussion of these factors (pages 3 and 4).

2. Regarding the results and discussion, you mention that results were only statistically significant for males and not females. The results in Appendix 3 are very interesting, but you barely discuss the implications. Why do you think model results by sex vary? Is this due to potential mechanisms that impact associations? I assume there are biological differences involved but it is difficult to tell, given a) I have little knowledge on the subject and b) there is very little information given regarding the analysis by sex– perhaps another figure, same as figure 2, giving the trends by sex could be useful. I suggest the results by sex should also be highlighted in the text, and the implications discussed in context. Also, it is not clear to me why you decided to focus more on the results stratified by age and not by sex, please elaborate on your decision. From what I understand you don't have data both by age and sex, which is another limitation that should be mentioned.

Thank you for these comments. We have now taken the analysis of differences by sex from the appendices and put it in the main paper (page 9, paragraph 3 and page 11, paragraph 3). Here we mention that the reasons for these potential differences are unknown but that it could be that women were more likely to be diagnosed pre-austerity, whereas the change in public spending may have been related to more additional cases among men. The lack of data by both age and sex was not available for this study due to the scale at which the data was aggregated and supplied, and we have now added this as a limitation in the limitations section (page 13, paragraph 1). We have also included a graph stratified by sex as figure 3 (mentioned in text: page 7, paragraph 1 and page 8, paragraph 3).

Other minor comments.

- In the data section, can you please clearly state the time period of analysis, and also note that LA annual figures are based on financial years.

Thank you for this comment. These details have been added in page 5, paragraph 2.

- Please use "sex" instead of "gender" when you mention stratified data for males and females.

Thank you for this suggestion. We have changed this throughout the manuscript

- Unfortunately, figure 3 is barely readable so I wasn't able to assess it properly.

Thank you for pointing this out. This figure has been removed. Reference to figure 3 now refers to the graph of change in admission rates over time stratified by sex.

VERSION 2 – REVIEW

REVIEWER	Sanabria, Marta Cristina Universidad Nacional de Asunción
REVIEW RETURNED	25-Jul-2022
GENERAL COMMENTS	The research is very interesting because it could have an impact on public policies related to nutritional anemias.

	It would be interesting to include the key words of the research and further highlight the objective in the abstract and after the discussion the main conclusions of the research.
REVIEWER	Alexiou, Alexandros University of Liverpool Faculty of Health and Life Sciences, Department of Public Health and Policy
REVIEW RETURNED	01-Aug-2022
GENERAL COMMENTS	The authors should be congratulated for their work in revising the manuscript, and addressing all the issues I raised so coherently.

VERSION 2 – AUTHOR RESPONSE

Reviewer: 2

Dr. Marta Cristina Sanabria, Universidad Nacional de Asunción

Comments to the Author:

The research is very interesting because it could have an impact on public policies related to nutritional anemias.

Thank you

It would be interesting to include the key words of the research and further highlight the objective in the abstract and after the discussion the main conclusions of the research.

Thank you. The key words we added on the system, which may not have been visible to the reviewer, were 'health policy', 'public health' and 'nutrition.'

In the abstract we have clarified the objective with the sentence – 'Specifically we address whether greater cuts to LA spending were linked to increased hospital admissions for nutritional anaemia.'

We thank the reviewer for pointing out that the original revision was lacking a conclusion section. We have now added this: 'Overall our analyses conclude that increased LA spending was associated with reduced hospital admissions for nutritional anaemia, with austerity having the opposite effect. This research adds to other accumulating evidence of the negative impacts of austerity and the importance of maintaining and increasing public sector spending.'

Reviewer: 3

Dr. Alexandros Alexiou, University of Liverpool Faculty of Health and Life Sciences

Comments to the Author:

The authors should be congratulated for their work in revising the manuscript, and addressing all the issues I raised so coherently.

Thank you